# Design of Acoustic Signal for Positioning of Smart Devices

**DOI:** 10.3390/s23187852

**Published:** 2023-09-13

**Authors:** Veronika Hromadova, Peter Brida, Juraj Machaj

**Affiliations:** Faculty of Electrical Engineering and Information Technology, University of Zilina, 01026 Zilina, Slovakia; veronika.hromadova@uniza.sk (V.H.); juraj.machaj@uniza.sk (J.M.)

**Keywords:** acoustic communication, smartphone, signal design, airborne channels, frequency band selection, positioning, indoor

## Abstract

This paper addresses the limitations of using smartphones in innovative localization systems based on audio signal processing, particularly in the frequency range of 18–22 kHz, due to the lack of technical specifications and noise characterization. We present a comprehensive study on signal design and performance analysis for acoustic communication in air ducts, focusing on signal propagation in indoor environments considering room acoustics and signal behavior. The research aims to determine optimal parameters, including the frequency band, signal types, signal length, pause duration, and sampling frequency, for the efficient transmission and reception of acoustic signals for commercial off-the-shelf (COST) devices. Factors like inter-symbol interference (ISI) and multiple access interference (MAI) that affect signal detection accuracy are considered. The measurements help define the frequency spectrum for common devices like smartphones, speakers, and sound cards. We propose a custom signal with specific properties and reasons for their selection, setting the signal length at 50 ms and a pause time of 5 ms to minimize overlap and interference between consecutive signals. The sampling rate is fixed at 48 kHz to maintain the required resolution for distinguishing individual signals in correlation-based signal processing.

## 1. Introduction

In the rapidly evolving world of smart devices, accurate positioning plays a crucial role in enhancing user experiences and enabling a wide range of applications. Unlike outdoor positioning, where GNSS (Global Navigation Satellite System) has become a standard technology, indoor positioning has yet to reach a consensus. However, GNSS does not work well indoors due to the high attenuation and complex reflections of electromagnetic waves in buildings. To determine the location of mobile devices in an indoor environment, it is possible to use data from various supporting technologies such as WiFi [1,2,3,4], Bluetooth [5], RFID (Radio Frequency IDentification) [6], ZigBee [7], MEMS (Micro-Electro-Mechanical Systems) sensors [8,9], UWB (Ultra-Wide Band) [10,11], geomagnetic field [12,13], LiFi (Light Fidelity) [14], and acoustic signals [15,16,17,18].

Acoustic positioning utilizes sound waves to estimate the position of devices and seems to be a promising solution. It offers several advantages such as accuracy at the centimeter level, compatibility with existing audio hardware, and the potential for implementation in indoor environments where GPS signals may be limited or unavailable. Acoustic positioning has found applications in various domains, including augmented reality, asset tracking, smart home automation, and location-based services. However, the accuracy and reliability of acoustic positioning heavily depend on the characteristics of the room in which the signal propagates. In recent years, significant advancements have been made in understanding and optimizing acoustic positioning techniques. This article aims to delve into the crucial design considerations for acoustic signals used in positioning. Signal design plays a pivotal role in achieving accurate and reliable positioning results. By examining various factors such as frequency selection, signal modulation techniques, timing considerations, noise mitigation, and signal encoding, we aim to provide insights into the optimization of acoustic positioning performance. To shed light on this important topic, a comprehensive series of measurements and experiments were conducted. This paper is focused on the quantification of the influence of room characteristics on signal propagation, particularly focusing on the reverberation time and its variations across different rooms. Moreover, the effect of different rooms on the size of the guard interval between signals, which can have implications for signal detection and decoding accuracy, was investigated. Through the analysis of the results from the measurements and their implications, valuable insights into optimizing acoustic positioning systems for different room environments are presented. Findings from the measurement campaigns contribute to the broader understanding of the challenges and opportunities associated with acoustic signal propagation in real-world scenarios. This comprehensive investigation contributes to the optimization of acoustic positioning by enabling effective deployment in diverse real-world scenarios. Understanding the design considerations can lead to the enhanced efficiency of acoustic positioning techniques.

We may categorize acoustic positioning systems into two fundamental groups based on the frequency range of the acoustic signal used for position estimation, namely, the frequency band in ultrasound, i.e., above 20 kHz [19,20] and the audible frequency band below 20 kHz [17,21]. The drawback of using frequencies below 20 kHz is that certain consumers may perceive their use as noise, which limits the appeal of such systems. Some systems seek a compromise as the use of high frequencies incurs additional costs for both mobile devices and infrastructure. Moreover, in implementation on smartphones, which have become an integral part of everyday life, the limitation is represented by the frequency response of the built-in microphone. Therefore in this work, acoustic localization systems are divided into three groups: ultrasound systems, i.e., above 20 kHz, audible sound systems, i.e., below 20 kHz, and systems that use a frequency band in the vicinity of 20 kHz [16,22,23,24,25,26,27]. The last group is the most interesting since there is the possibility to use mobile phones for localization purposes, while frequencies in the vicinity of 20 kHz are inaudible for the vast majority of users [28,29].

The number of localization systems based on audible acoustic signals is limited since signals from such systems would distract the users, who could perceive the signals as noise. On the other hand, systems that use signals above 20 kHz are gaining some attention. The frequency band is located above the hearing limit; therefore, it does not disturb users in the area. However, the use of such frequencies comes with a disadvantage: the need for specialized technology that works with high frequencies, which affects the cost of creating such a system. Another limitation is the need for a high sampling frequency, as the Nyquist criterion must be obeyed. Some localization systems use a frequency band at the limit of hearing, i.e., close to the frequency of 20 kHz. The main advantage is that such a frequency spectrum is available for use in smartphones and it is, thus, possible to reduce the cost of system deployment.

The real-time application of acoustic signals for smart device positioning offers numerous practical and innovative opportunities across various fields. It enables precise indoor localization, asset tracking, context-aware services, emergency response, and smart home automation. Additionally, it enriches augmented reality experiences, facilitates proximity-based interactions, and supports location-based marketing. The versatility and potential of acoustic positioning in enhancing user experiences, optimizing processes, and enabling innovative services make it a promising technology with a bright future ahead.

Acoustic positioning offers several advantages in various applications. Firstly, it provides high precision, particularly in environments with favorable signal conditions, enabling accurate positioning of smart devices. Secondly, acoustic positioning systems have low infrastructure requirements, as they can be implemented using existing devices or readily available speakers and microphones. This makes it a cost-effective solution for many applications. Additionally, acoustic waves can propagate around obstacles, granting the system non-line-of-sight capability, which is advantageous for positioning in obstructed environments. Lastly, acoustic signals are well suited for indoor positioning applications since they do not penetrate walls. Various parameters can be measured and then used to estimate the position, such as received signal strength (RSS), time parameters (time of arrival—ToA, time difference of arrival—TDoA, etc.), direction parameters (angle of arrival—AoA, direction of arrival—DoA), phase parameters (phase of arrival—PoA, the phase difference of arrival—PDoA). Among the main ones is the level of the received signal, which is very often used in systems working with a radio signal (RS). For an acoustic signal, a time parameter and its variants are very often used. The position estimation is performed based on the measured parameters that were mentioned above. In Table 1, there is an overview of the basic methods for determining the location [15].

There are certain limitations to consider with acoustic positioning. Signal interference from background noise and other acoustic sources can impact the accuracy of the positioning system. Careful noise reduction techniques and signal processing algorithms are necessary to mitigate this interference. Acoustic signals also have a limited range compared to other wireless technologies, which may restrict their use in larger spaces. Environmental factors such as temperature, humidity, and air density can affect the speed and propagation of sound waves, potentially impacting positioning accuracy. Therefore, accounting for these factors and implementing appropriate calibration techniques is crucial to ensure reliable and precise results from the acoustic positioning system.

## 2. Materials and Methods

When designing an acoustic signal, we must focus on the following aspects: the frequency spectrum, interference caused by multiple access, signal modulation, signal duration, noise, and interference mitigation. The frequency spectrum for the signal design must be chosen with consideration for the devices for which it is intended, and other criteria, such as the inaudibility of the sound in our case. In this way, the signal does not disturb the users. Because acoustic systems use various sources to calculate position, the manner of access to the communication channel must be considered. Different signal modulations are employed to improve the efficiency and channel width utilization. Consideration of the signal duration is critical for ensuring optimal performance while conserving energy and resources. Assessing the degree of noise in the specified spectrum and identifying its sources aids in calculating the SPL of the signal and the approaches for suppressing it.

### 2.1. Frequency Selection for the Acoustic Signal

The lack of technical specifications for mobile phones limits their potential for use in new localization systems based on the use of audio signal processing. One of the parameters is frequency sensitivity—frequency response. Our main goal was to identify the sensitivity in the upper limit of the audible range. Acoustic frequencies between 20 Hz and 20 kHz are audible to the human ear. The ability to hear the higher frequency, on the other hand, is impaired. According to the available sources, people over 18 cannot hear a frequency above 18 kHz [28,29]. According to studies on the effects of ultra-sonography on human hearing, an acoustic signal with a frequency outside the hearing range played on common computer hardware does not pose any health risks [28,30]. We examined common mobile phones, and the resulting frequency response curves show that all Android smartphones have a reasonably flat frequency response up to 21.2 kHz. A significant decrease in sensitivity only occurs above this frequency. The evaluation of iPhone brand smartphones showed a slightly worse frequency range, with a sharp drop above 20.6 kHz. This drop may have occurred due to anti-aliasing filtering, as the sample rate in the built-in A/D converter was set to 44.1 kHz. If a higher sampling rate is used, the effect of anti-aliasing filtering can certainly become less significant (note that 44.1 kHz sampling was used to maintain the same recording conditions for all tested smartphones). Measurements were made to prove that all tested smartphones are suitable for inclusion in sound-based localization systems that operate at frequencies above the audible range. A detailed description of the measurement can be found in the publication [31].

### 2.2. Interference Caused by Multiple Access

Access to the acoustic communication channel must be considered while employing an acoustic signal for positioning. To prevent collisions, the acoustic signal is frequently broadcast from static reference nodes using a TDMA (Time Division Multiple Access) strategy. While N reference nodes and loudspeakers emit the signal *g**_j_*(*t*), the microphone ms receives the signal *r*(*t*) [32,33]:(1)$$r\left(t\right)=\sum_{j=1}^{N}A_{j}\cdot \left(h_{j}*g_{j}\right)\left(t-t_{j}\right)+n\left(t\right),$$
where *t_j_* and *A_j_* are ToA and signal amplitude, *n*(*t*) represents the channel noise, and *h**_j_*(*t*) represents the previously unknown impulse response. The effect on the signal *g**_j_*(*t*) is described by the convolution *h_j_*(*t*)∗*g_j_*(*t*).

The cross-correlation of *g_k_(t)* and *g_j_(t)* codes is described by *R_gkgj_*(*t*), and *η*(*t*) represents convoluted noise. The output of the receiver is formed by correlating *r*(*t*) with all signal codes. Then, for the *k*-th receiver, we obtain:(2)$$R_{rg_{k}}\left(t\right)=A_{k}\cdot \underset{ISI_{k}}{\underbrace{\left(h_{k}*R_{g_{k}g_{k}}\right)\left(t-t_{k}\right)}}+\underset{MAI_{k}}{\underbrace{\underset{j\neq k}{\sum }A_{j}\cdot \left(h_{j}*R_{g_{k}g_{j}}\right)\left(t-t_{j}\right)}}+\eta \left(t\right).$$

As can be seen from Equation (2), there are two strong influences on the accuracy of the estimation ToA based on cross-correlation: one is between symbol interference ISI, i.e., the first part of the equation, which arises due to the limited width of the acoustic channel and reduces the correlation local maxima, thus degrading signal detection; another is MAI, i.e., the second part of the equation, among all the resulting codes, where signals with a larger amplitude make it difficult to detect weaker signals transmitted at the same time. The combination of ISI and MAI can lead to large deviations in ToA and TDoA estimates from the actual values. In [34], the authors compensated for this effect using recursive subtraction techniques, namely, the parallel interference algorithm; however, this compensation introduced an additional time delay to the system. 

The system setup: Carrier frequency—16 kHz; bandwidth—8 kHz; BPSK modulation; 63-bit Kasami-coded signal; sampling frequency—96 kHz. All transmitters were located on one wall at different heights. They made measurements in one room with dimensions of 3 × 3 m, where an area next to the wall with a size of 1 × 1 m was chosen and divided it into 25 points. The measurements of the effect of the noise were conducted here, wherein white Gaussian noise was produced and the effect was monitored on the recognizability of the signals. Emissions with SNRs of 12, 9, 6, 3, and 0 dB were considered. As the SNR of the emitted signals decreased, the system availability of those points affected by MAI also decreased and the corresponding positioning error increased. Conversely, in those test points where good results were obtained in the absence of noise, the system performance was more robust against a decrease in the SNR.

A different approach was used in [35], where they used signal coding to recover the time multiplex, which helps to identify the transmitter. Another way to avoid MAI is to employ a different multiple access technique, such as TDMA, wherein each emitter is allocated a specific time slot to utilize the channel, effectively reducing the chances of signal overlap. However, the drawback of TDMA is its relatively slower positioning rate, mainly due to the slower speed of acoustic waves. To address this limitation, an alternative solution called T-CDMA (an intermediate between TDMA and CDMA) was proposed [36]. The concept behind T-CDMA involves introducing a specific delay between emissions, which helps mitigate the superposition of signals while still maintaining the benefits of CDMA separation, as some level of signal overlap in the channel persists.

### 2.3. Signal Modulation Techniques for Acoustic Signals

To increase the effective use of the channel width, two approaches are mainly used in signal modulation, namely, linear frequency modulation LFM (Linear Frequency Modulation) and BPC (Binary Phase Coding). In the first case, a change in frequency is used, namely, its linear increase or decrease from *f*_1_ to *f*_2_ during the duration of the pulse:(3)$$f=f_{1}+k\cdot t,$$
(4)$$x\left(t\right)=\cos\left(2\pi f_{1}t+k\pi t^{2}\right),$$
where *k* represents the rate of frequency change, 0 ≤ *t* ≤ *τ*, and *τ* is the duration of the pulse. In the second case, the long pulse is divided into *N* sections, the phase of which is set to 0 or *π* radians, according to the given bit in the code. Using a pseudo-random sequence as a code, the waveform approximates a noise-modulated signal. When the time parameters are measured, the signal reception time is determined when the auto-correlation local maximum exceeds the threshold value. The main advantage is that, when generating different sequences from the same group, these sequences have almost zero cross-correlation. Therefore, when simultaneously broadcasting by different sources, there is only a small interference between them. The most frequently used pseudo-random sequences are Kasami [35], LS [37], Gold [38], and CSS [27]. This approach is suitable when there is a wide frequency band available.

### 2.4. Signal Duration and Timing Considerations

Signal duration and timing play crucial roles in acoustic positioning systems. The duration of the signal refers to the length of time the signal is transmitted, while timing refers to the precise timing of signal transmission and reception. The duration of the signal should be carefully determined to balance between achieving accurate positioning and minimizing power consumption. A longer signal duration allows for better signal detection and improved positioning accuracy, but it may result in increased power consumption. Moreover, there is a trade-off with system responsiveness: shorter signal durations can enhance system responsiveness by allowing faster updates of position estimates. However, shorter signals may be more susceptible to noise and may require sophisticated signal processing techniques to extract accurate position information. The authors in [39] use a 50 ms signal length, while others use 36 ms [40], 40 ms [24], or 110 ms [27].

In the case of acoustic signals, cross-correlation is often used to compare the mutual time delays between these signals [41]. In this way, it is possible to determine the difference in signal arrival times and, thereby, locate the sound source in space. The cross-correlation for the signals is defined as follows:(5)$$C\left(\tau \right)=\int_{0}^{T}y\left(t\right)\cdot Ref\left(t-\tau \right)dt,$$
where *C*(*τ*) is the correlation between the signals *y*(*τ*) and reference signal *Ref*(*τ*) at time *τ*, and *y*(*t*) and *Ref*(*t* − *τ*) are waveforms of signals shifted in time by *τ*. Cross-correlation provides information about the similarity of signals at different time shifts. If the cross-correlation is high at a particular time offset, it indicates that the signals have similar patterns and the time difference between them is given by that offset. In the case of sound localization through cross-correlation, the TDoA between the arrivals of individual signals is used. Cross-correlation between the signals makes it possible to determine this time difference, which is necessary for the subsequent localization of the sound using hyperbolic trilateration or other methods. 

It is necessary to choose the design of the signal used for localization appropriately. The frequency range of 18–20.6 kHz was selected for two main reasons. First, it falls above the 18 kHz threshold [28,29], making the acoustic signal inaudible for most humans. Second, it aligns with the frequency characteristics of the microphones, as explained in Section 2.1. and detailed in our publication [31]. This frequency range provides us with 2.6 kHz of bandwidth; thus, we needed to create four distinct signals, each designed to be easily distinguishable through correlation. With these considerations, we ensured efficient signal separation and accurate identification of the acoustic signal sources. We set the signal length to 50 ms based on the results of the authors in [42], who found that a signal with a length of 40 ms or more has an 80% higher rate efficiency (number of successful experiments/total number of experiments) and the average error of the estimated distance is lower by 0.05 m. Furthermore, the signal frequency is constrained by the array spacing, which must fulfil Equation (6):(6)$$d<\frac{c}{2B},$$
where *d* is the length of the signal, *c* is the speed of sound, and *B* is the frequency band, which means that the length of the signal has to be shorter than 0.066 s.

ISI and MAI influenced our decision-making, and we chose the frequency separation of unique signals and LFM, which enabled us to determine the signal’s source using correlation. The description of the used signal can be found in Table 2. We tested the individual signals from the point of view of auto-correlation and cross-correlation with each other in ideal conditions and after passing through a real environment. 

In the experiments, we used a sampling frequency of 48 kHz. We will gradually compare individual signals in ideal and real conditions. In Figure 1, there is a corresponding signal in each row, i.e., in the 1st row, it is signal 1, likewise in the column. On the main diagonal, there is the auto-correlation for each signal, i.e., S1 with S1, S2 with S2, etc. As we can see, the signal was well designed for an ideal environment and there are significant maximums.

To find out how the signal was distinguishable after passing through a real environment, we recorded one type of signal repeated five times and then we verified the discriminability of the signals by the correlation on these recordings. The results of the experiment can be found in Figure 2. A sampling frequency of 48 kHz was sufficient to distinguish individual signals after passing through the acoustic channel.

### 2.5. Signal Encoding and Decoding

Various encoding techniques are employed in acoustic positioning to embed position data into the acoustic signal. Data can be added to an audio signal using audio signal processing through a process called modulation. Frequency-shift keying (FSK) is a commonly used method to encode data into sound. In binary FSK, two frequencies are used, while quadrature FSK uses four frequencies. To increase the amount of data transmitted in a specific time frame, phase-shift keying (PSK) can be used. PSK achieves more efficient data transmission by utilizing changes in the signal’s phase. Through these techniques, data can be effectively embedded and transmitted within an audio signal. These encoding techniques ensure that the transmitted signal carries the necessary data for accurate position estimation. 

On the receiving end, decoding algorithms are employed to extract the position information from the received signals. These algorithms analyze the received signal characteristics, such as the frequency, phase, or time of arrival, to determine the position-related data embedded in the signal. The choice of decoding algorithm depends on the encoding technique used and the specific requirements of the positioning system. Error correction techniques are often applied in the decoding process to enhance the robustness and reliability of position estimation. These techniques help mitigate the effects of signal degradation, noise, and interference. Robustness considerations are also vital in signal encoding and decoding. Positioning systems should be designed to withstand environmental variations, signal degradation, and interference. Robust encoding and decoding techniques, combined with error correction mechanisms, can enhance the system’s resistance to noise and improve the overall positioning accuracy. 

Overall, signal encoding and decoding are critical components of acoustic positioning systems. By employing suitable encoding techniques, implementing effective decoding algorithms, and considering error correction and robustness considerations, accurate and reliable position estimation can be achieved in various applications.

### 2.6. Noise and Interference Mitigation

Noise and interference mitigation is a critical aspect of acoustic positioning systems to ensure accurate and reliable position estimation. Various techniques and strategies are employed to minimize the impact of ambient noise and interference on the received signals. A comprehensive analysis of ambient noise helps identify its frequency range, intensity, and temporal variations. This analysis aids in designing effective mitigation strategies. Identifying the sources of noise, such as background chatter, machinery, or environmental factors, assists in implementing targeted noise reduction techniques. These findings assist us in selecting the right frequency range for the acoustic signal to reduce interference from background noise and ensure adequate signal detection by localization devices. Our main attention was on the impact of the equipment in the room and personnel in offices and classrooms.

#### 2.6.1. Description of Background Noise Measurement

For the measurement, we selected different rooms and conditions to cover various scenarios: a room without any computing equipment and people, the same room with people present, an office space with and without people while PCs were running, with and without carpet, a department store with people during a lunch break, and for comparison, an outdoor space near a road and a recording from a loud metal group performance. We used a Xiaomi Redmi Note 10 Pro recording device with a sampling frequency of 48 kHz. In each room, we recorded five sessions, each lasting for 1 min. After recording, we filtered out all frequencies except for the 18–20 kHz range. The data were then normalized based on dB values, using the Norsonic Nor140 acoustic analyzer class 1 of sound analyzers. We conducted a level analysis using the analyzer, obtaining *L_eq_* values representing the time average pressure, and calculated their differences. The data are related to 0 dB, which corresponds to the acoustic threshold of hearing, equivalent to 10^−12^ Wm^−2^. A photo from the measurement can be seen in Figure 3.

#### 2.6.2. Measurement Results of Background Noise in the Frequency 18–20 kHz

Average values from measured SPLs for individual conditions and rooms can be found in Table 3. For better visual clarity, we also present the data in Figure 4.

The precision of the measurement based on the specification of the Norsonic Nor140 was a gain accuracy at 1 kHz: ±0.2 dB, and a frequency response re. 1 kHz: ±0.5 dB for 20 Hz < f < 20 kHz. The background noise in the quiet classroom averaged −18.0 dB SPL. In the presence of lights, the value did not change, which indicates no influence of lights on the noise level in this frequency range. With the presence of 15 people in the room, the background noise value increased by 2.7 dB. As the number of people in the room increased to 25 and 37, the SPL values further increased by 3.1 dB SPL and 4.4 dB. It should be noted that these recordings were made when students entered the room and then sat down and made noise with chairs and tables. The obtained data indicate that such a presence of people had a greater effect on the background noise level than turning on the lights. In the office with carpet, an average of −18.0 dB SPL was recorded. The presence of four people in the office and the use of a PC did not increase the value. In this office, there was a carpet, so it dampened the reflections from the floor. On the other hand, in the room office without carpet with the presence of four people, an average of −17.0 dB SPL was recorded. Therefore, there was an observable increase in the noise level compared to the carpeted room. An average of −17.1 dB SPL was recorded in the classroom with computers, where the server was located, so we expected a higher noise level compared to the previous rooms without people. The presence of 15 people in the room and turning on 15 PCs again caused an increase in the value by 1.5 dB. The average value of the background noise in the frequency of 18–20 kHz in the shopping center during lunch, where the recording was made at food outlets with the presence of more than 100 people, reached −14.6 dB SPL. For comparison, the road outside was noisier at −13.6 dB SPL, compared to the indoor measurements. An extreme case was the measurement during the rehearsal of a metal group, where the noise reached the value of −5.7 dB SPL. This significant increase in noise level was probably due to the loud performance of the metal band, which is typical of this music genre. To provide a clear sample and illustrate the nature of noise in this frequency range, we included spectrograms from the shopping center and a rehearsal of the metal band, see Figure 5. In indoor environments, noise in this frequency spectrum is predominantly caused by human activity.

The key finding from this measurement of the background noise is that the SPL in the selected frequency range (18–20 kHz) was relatively low, indicating minimal interference with the acoustic signal utilized for position estimation. Our measurement brings a novel contribution as we could not find any existing scientific articles that focus on noise measurement in this particular frequency spectrum.

### 2.7. Influence of Room Characteristics on Signal Propagation 

The characteristics of a room have a significant influence on the propagation of acoustic signals within it. The room dimensions, shape, and surface materials play a crucial role in determining how sound waves travel and interact within space. The size of the room affects the resonant frequencies and modes of the acoustic waves. Larger rooms tend to have lower resonant frequencies, while smaller rooms exhibit higher resonant frequencies. These resonances can result in the amplification or attenuation of specific frequencies, leading to variations in the perceived sound and affecting the accuracy of acoustic positioning. The shape of the room can cause sound wave reflections and diffraction. Irregularly shaped rooms may introduce multiple reflections and the scattering of sound waves, resulting in complex interference patterns. The surface materials of the room, such as walls, floors, and ceilings, the presence of furniture, objects, and the occupants within the room, can further influence the absorption, reflection, and diffusion of sound. Highly reflective surfaces, like glass or polished metal, can cause strong reflections, while absorbent materials, like carpets or acoustic panels, can reduce reflections and reverberations. 

We have divided the investigation of these influences into the definition of spaces, which we will describe by their properties and the acoustic parameter reverberation time. In the next section, we will describe how different rooms and the size of the guard interval affect the accuracy of ToA determination. 

#### 2.7.1. Room Impulse Response

Room impulse response is the characteristic response of a room to a short acoustic impulse—a Dirac impulse. The behavior of a linear and time-invariant system can be obtained by convolving the input signal with an impulse response. Assuming that the setup of the acoustic signal source and the microphone is stationary, the sound propagation and reflections in the room can be considered a close approximation of a linear and time-invariant system [43]. This response describes how sound waves travel in a room and how they bounce off walls, floors, ceilings, and objects. The impulse response of the room is important for the acoustic design of the room because it determines how the noises in the room will sound, e.g., amplified or muffled. From a mathematical point of view, the impulse response of the room is represented as the time response of the room, which is obtained from the measurements of the acoustic signals. Measurements are made using a sound source that emits a short sound pulse and a microphone that captures the room’s response to this pulse. The room response is then analyzed using mathematical tools such as the Fourier transform, which decomposes the audio signal into its frequency components. The result of the analysis of the impulse response of the room is the impulse characteristic of the room, which is represented by a graph that shows how the amplitude of the acoustic signal changes with time.

#### 2.7.2. Description of Reverberation Time Measurement 

We conducted the measurement of room reverberation time following the ISO 3382-1 standard [44], which offers guidance and procedures for assessing the acoustic characteristics of various spaces, including rooms within buildings. This standard focuses on the characterization of rooms using several parameters, with the reverberation time being the most well known. Reverberation time denoted as *T60* refers to the duration it takes for the sound pressure level (SPL) to decay by 60 dB. In most contexts, a decline of 60 dB is difficult to accomplish. T20 or T30 was used and extrapolated to determine the reverberation time. This is a good approximation because the sound pressure level during decay is regarded as linear in the logarithmic scale. *T20* represents the time it takes for the energy of the RIR to decrease from −5 dB to −25 dB. The Nor140 calculated the T20 value, which was normalized to the required 60 dB decay time.
(7)$$T20=E^{-1}\left(-25\right)-E^{-1}\left(-5\right),$$
where *E*^−1^(*ξ*) corresponds to the time delay *t_ξ_*, for which *E*(*t_ξ_*) = *ξ*; in simpler terms, ξ represents the SPL value (−5 dB or −25 dB). The room impulse response measurement process consists of the following steps:Preparation of the measuring device: sound source, in our case balloons inflated to a diameter of 30 cm, and microphone for capturing the acoustic signal: acoustic signal Norsonic Nor140 analyzer, calibrated according to the standard;Location of the measurement system: The microphones were placed in a pre-defined position in the room. The sound source was also placed in a pre-determined position, usually near one wall of the room;Generation of an acoustic impulse: The sound source generated a short sound impulse that spread through the room, in our case a balloon burst. Sounds close to the Dirac impulse were used (balloon burst, a stun gun shot) or sounds were played from omnidirectional speakers, e.g., a scattered sine, a sequence with a maximum length—MLS;Impulse response recording on the Norsonic Nor140 analyzer;Data processing and analysis: Recorded data were processed and analyzed using various algorithms and techniques. For example, the Fourier transformation was performed, which allowed the audio signal to be decomposed into frequency components. Based on these data, various room acoustic parameters such as reverberation time were calculated.

We selected five rooms based on their size and diversity to find out exactly how this diversity manifested itself in the reverberation time impulse response of the room. Table 4 provides a description of the rooms chosen for measuring the room’s reverberation time. Every room is described by its size and room equipment. We wanted to establish their acoustical properties and compare them, and then utilized the same rooms to analyze the impact of the guard interval between signals on the ToA estimation error. The selection criteria were their diversity in terms of size and expected acoustic properties.

In each of the selected rooms, we placed the acoustic signal analyzer in the center of the room at a height of 1.1 m above the ground. Then, we burst the balloons in four places. The time between bursts was more than 10 s, so the acoustic signal had time to completely disappear. We can see the layout of the measurement in Figure 6.

#### 2.7.3. Measurement Results of Reverberation Time

The Norsonic acoustic analyzer has an automatic evaluation of the T20 parameter for frequencies from 50 Hz to 10 kHz. The measurements were only up to a value of 10 kHz, as our Norsonic Nor140 equipment is limited to this value, and various software for determining these parameters are normally limited to 8 kHz or up to 10 kHz. These limits are because higher frequencies are uninteresting from an acoustic point of view (music, speech). The average of the measured values can be found in both Table 5 and in Figure 7.

From the measured values, we can see the effect of the room on the impulse response and, thus, the speed of attenuation of the acoustic signal. Since the values we are interested in are outside the measurement range, we can assume from the trend of the curve in the graph that this value gradually decreased for all types of rooms. On each curve, we see the maximum and, thus, the frequency that was amplified in the given room. This frequency is also called the room’s mode. Parameters of A-netw. and Z-netw. are their averages determined on the frequency spectrum of the measurement. Z-netw. represents real measured values, and A-netw. is adapted to the human hearing curve. By observing the effect of frequency on *T20* values, we found that lower frequencies had longer reverberation times (larger *T20* values), which means that these frequencies in the room were reflected longer and had more acoustic energy. Conversely, higher frequencies had shorter reverberation times (smaller *T20* values), indicating faster attenuation and less acoustic energy at these frequencies. The assumption for the decreasing nature of the curve is based on the properties of the acoustic signal, i.e., attenuation by propagation, and higher frequencies are attenuated faster based on ISO 9613-1 [45] and ISO 9613-2 [46].

Another factor we considered is the type of room. We found that the small office achieved the lowest *T20* values for most frequencies. This room was the smallest and had one bare wall, while the other walls were covered with cabinets or bookcases. This configuration resulted in more dispersion and sound attenuation in the room of the small office, which explains its lower *T20* values. On the contrary, the highest values were measured in the corridor, where all surfaces except the ceiling were highly reflective, as can also be seen on the course of the curve. From the value of 4 kHz, all the curves had a similar course, decreasing as the higher frequencies were damped more quickly due to the influence of propagation.

In summary, it can be concluded that the Norsonic acoustic analyzer provided important information about the reverberation time for different frequencies in the analyzed rooms. This information allowed us to evaluate the acoustic properties of rooms and their ability to maintain acoustic energy at different frequencies. We found that the room size, layout, materials, and finishes had a significant impact on these characteristics. Based on the analysis, we noted that the *T20* values changed with frequency and reflected the rate of sound attenuation in the room. Lower frequencies tended to be sustained longer and exhibited more acoustic energy, while higher frequencies had a faster decay and less acoustic energy.

#### 2.7.4. The effect of Different Rooms on the Length of the Guard Interval

To investigate the impact of various room characteristics on the length of the guard interval between signals in acoustic positioning systems, we designed and implemented a measurement setup. The primary goal of this experiment was to understand how room properties influence the necessary guard interval between successive signals to achieve accurate positioning. The guard interval is essential in compensating for ISI, which affects the correlation maxima, making it challenging to precisely determine the ToA of the transmitted signal and introduce errors. Measurements were performed in different rooms with diverse acoustic properties, representing a range of sizes, shapes, and materials, as described in Section 2.6.2. The emitters emitted signals at specific intervals, and the receivers recorded them. During the measurements, the guard interval between signals was systematically varied, starting from a minimal value and gradually increasing it. By analyzing the received signals, we assessed how room parameters, such as size, shape, surface materials, and reverberation time, affected ToA based on the size of the guard interval between signals. These factors can influence signal propagation, reflections, and reverberation within the room. The list of rooms can be found in Table 4. Logitech Z200 speakers were distributed as acoustic signal sources in each room, which were connected to an external sound card and a PC. The receiving device was the Xiaomi Redmi Note 10 Pro smartphone, which was part of the tests on the frequency response of the microphones [31]. More detailed specifications of devices can be found in Table 6. 

#### 2.7.5. Measurement of the Acoustic Signal Source

The first measurement was the frequency response of the sound source, i.e., Logitech Z200 speakers, to determine the usability of the speakers as acoustic signal sources. We used a calibrated Norsonic Nor140 sound level meter as a reference measuring device. Figure 8 shows the measured frequency response of the loudspeakers.

The SPL values were normalized to the SPL value at a frequency of 15 kHz. It can be seen from the frequency characteristic that their use in the required range of 18–20.6 kHz was possible, as the greater decrease only occurred after exceeding the frequency of 20.8 kHz.

#### 2.7.6. Description of Measuring the Effect of Different Rooms on the Length of the Guard Interval

Individual speakers were placed in the corners of the room under the ceiling. They were connected to an external sound card and to a PC, which ensured the synchronization of the speakers. In each room, we selected five places for measurements: the first was the center of the room and the rest were 1 m from the wall also in the center. The scheme of signal transmission and the location of the microphone can be seen in Figure 9. Via measuring, we tested the appropriate length of the pause between signals in different spaces. Four signals are described in Table 2, one for each speaker. We started by sending all four signals at once. Then, we sent signals without a gap, i.e., immediately followed by a pause of 0 ms. Subsequently, we sent signals with a step of 0.5 ms up to 10 ms. We created 22 sequences. The guard intervals were set as follows: the first sequence contained all four signals that were transmitted at once, and the next sequence contained all four signals that were transmitted one after the other and, thus, the pause was 0 ms. The next sequences again contained all four signals, but the pause between them gradually increased with a step of 0.5 ms to a value of 10 ms. In each room, we measured 50 repetitions for each of the 22 sequences at a selected location. We recorded the transmitted signal and then processed the recordings. Based on the correlation, ToA was obtained from the recordings, and errors in ToA determination were subsequently calculated. Statistical processing of these error values was performed for each room and different guard interval lengths using the MATLAB environment and statistical toolbox. The goal was to analyze the error distribution and determine its parameters. By examining the data, we aimed to identify the guard interval length that minimizes the error in ToA estimation, providing valuable insights into optimizing the acoustic positioning system for accurate and reliable performance. For more clarity, see Figure 10.

#### 2.7.7. Measurement Results

ToA was computed from the recordings based on the correlation, and errors in ToA determination were then calculated. Using the MATLAB environment and statistical toolbox, these error values were statistically processed for each room and each guard’s interval lengths. The purpose was to determine the parameters of the error distribution. We tried to identify the guard interval length that minimized error in the ToA estimate by evaluating the data. We discovered that the lognormal distribution best described the data. The lognormal distribution is a continuous-type probability distribution used to model random variables that have logarithmic transformations with a normal distribution. This type of distribution is characterized by two parameters: the parameter mi (*µ*), which represents the mean value, and the parameter sigma (*σ*), which expresses the standard deviation of values [41]. 

We conducted analyses for every space and every pause length. The values that resulted are shown in Figure 11. In contrast to the much inferior outcomes that were anticipated given that it was a mechanical wave and the addition of the acoustic signals, no significant variation was observed when broadcasting all the signals at once compared to the other pause sizes. The values changed slightly during a value of 0 ms, but the errors in the signals’ arrival times remained almost the same. The value steadily changed and the error in the signals’ arrival times decreased as the pause size rose from 0.5 ms to 5 ms. The smallest dispersion of values and the smallest error in the signal arrival times were observed with a pause size of 5 ms. With a pause longer than 5 ms, the value of the mean error gradually changed and the error in the time of signal arrivals increased.

Measurements show that the value of μ decreased slightly towards the zero axis for a guard interval equal to 5 ms. This indicates that a guard interval of 5 ms represents the smallest error in ToA estimation for the selected signal design. We expected the course of the curve to show an improving trend with an increasing protection interval. The data show that the room did not have a great influence on the accuracy of Time of Arrival (ToA) estimates. Errors in ToA of around 10 microseconds, with the speed of sound (c) being 343 m/s, resulted in position estimation errors on the order of centimeters. The value seems to depend on the shaping of the audio signal and the used frequency band of the audio signal and sampling rate. The outcome was affected by the aliasing and artefacts resulting from the sampling process.

Therefore, the results of this specific measurement may not be universal and may depend on the specific signal design and hardware used. It is necessary to consider these factors and perform thorough measurements for any other specific signal designs. 

### 2.8. Guidelines for Designing Acoustic Signals for Localization Purposes

The steps for the design of an acoustic signal for localization purposes, in general, can be seen in Figure 12. 

Individual design steps:Objective definition: Clearly define the objective of the acoustic localization system. Identify the specific localization requirements, such as accuracy and range. Consider environmental conditions such as room dimensions, materials, and ambient noise. Consider the limitations and compatibility with the hardware and software capabilities of the target devices, such as smartphones;Frequency band selection: Choose an appropriate frequency band for the acoustic signal design. Consider the trade-offs between higher frequencies (better resolution but shorter range) and lower frequencies (lower resolution but longer range). Consider target device limitations;Signal parameters: Decide on the type of acoustic signal to be used, such as a continuous wave, chirp signals, or coded signals (e.g., pseudo-noise codes). Determine the essential signal parameters, including signal duration, modulation scheme (e.g., BPSK, FSK), and coding method (e.g., Kasami codes). Choose a suitable sampling frequency for signal processing. Higher sampling rates provide better resolution but may require more processing power;Guard interval: Calculate and set an appropriate guard interval between consecutive signals to avoid interference and ensure accurate signal detection. This interval should consider the signal’s duration, room acoustics, and potential signal reflections;Signal design validation: Use simulation or analytical tools to validate the design and ensure it meets the desired localization requirements. Test the signal in a controlled environment to verify its performance. Conduct real-world experiments in different environments to evaluate the acoustic signal’s effectiveness and accuracy for localization;Iterative optimization: continuously refine and optimize the signal design based on experimental results and feedback from real-world tests.

In summary, designing an effective acoustic signal for localization requires a systematic approach involving clear objective definition, careful frequency band selection, the determination of signal parameters, an appropriate guard interval calculation, thorough signal design validation through simulations and experiments, and continuous iterative optimization. By following these guidelines, developers can create acoustic signals that meet specific localization requirements and are compatible with the hardware and software capabilities of target devices.

## 3. Results and Discussion

Based on our measurements of the frequency response of the microphones of various smartphone models, we conducted an analysis to determine the appropriate frequency band for our acoustic communication system. We aimed to select a frequency range that would be effective in transmitting and receiving signals while utilizing the capabilities of the available hardware. To achieve this, we utilized an acoustic signal with frequencies outside the hearing range, specifically above 18 kHz. Through our measurements and analysis, we determined that the frequency band of 18–20.6 kHz would be suitable for our purposes. This provided us with a bandwidth of 2.6 kHz, which we could utilize to create distinct signals for differentiation. Considering that we had four sound sources available, we designed four types of signals to facilitate signal identification. It is important to note that acoustic signals, being mechanical waves, are susceptible to ISI and MAI. These factors can impact the accuracy of signal detection and correlation-based search methods. 

To mitigate these effects, we employed both up-chirp and down-chirp signals in our design. The selection of signal length played a crucial role in optimizing the performance of our system. Based on the findings of previous authors [42], who investigated the relationship between signal length and effectiveness, we set the signal duration to 50 ms. Their research revealed that signals with a length of 50 ms or more exhibited an 85% higher effective rate and a lower average distance error of 0.05 m. This informed our decision to set the signal length at 50 ms, ensuring a robust and accurate signal transmission. 

Furthermore, we considered the duration of the pause between consecutive signals. Through our measurements, we examined the impact of the pause duration on measurement errors in ToA. After a careful analysis, we determined that a pause length of 5 ms was be appropriate for our system. This duration allowed for sufficient separation of individual signals in both the time and frequency domains, minimizing the potential for overlap or interference. To visualize the resulting signal design and its parameters, we present Figure 13, which provides a clear representation of the selected frequency band, signal types, signal length, and pause duration. This graphical representation offers a comprehensive overview of our signal design approach. Lastly, we set the sampling frequency to 48 kHz for our experiments involving correlation-based signal processing. 

Lower frequencies were found to be inadequate in ensuring the necessary resolution for distinguishing individual signals. Therefore, we opted for a higher sampling frequency to maintain the desired level of precision and accuracy in our system. By carefully considering these factors and implementing the appropriate design choices, we established a robust and efficient acoustic communication system. Our signal design incorporates the selected frequency band, distinct signal types, optimal signal length, and guard interval.

In the comparison of the listed acoustic signal designs in Table 7, several key characteristics stand out. Each signal is tailored to suit specific requirements and environments, showcasing the versatility of acoustic positioning systems. It is important to note that none of the listed designs included measurements to determine the guard interval or background noise. The focus of the comparison was on signal design in existing acoustic positioning systems.

## 4. Conclusions

In conclusion, this comprehensive study addresses the limitations posed by the lack of technical specifications for COST devices and knowledge about noise characterization in the 18–20 kHz frequency range, which obstructs the utilization of smartphones in innovative audio-based localization systems. The research presents a detailed investigation into the signal design and performance analysis for acoustic communication in airborne channels, with a specific focus on signal propagation in indoor environments while considering room acoustics and signal behavior.

By determining the optimal parameters for signal transmission and reception, including the use of both up-chirp and down-chirp signals to mitigate inter-symbol interference and multiple access interference, the study enhances our understanding of effective acoustic communication systems. The chosen signal length of 50 ms and pause duration of 5 ms were designed to minimize the error of ToA measurements. Moreover, the sampling frequency of 48 kHz provides sufficient resolution for accurate signal detection using correlation-based signal processing techniques. The extensive and numerous measurements conducted were specifically customized for COST devices, offering a comprehensive understanding of the limitations and key factors influencing acoustic signal design. These results emphasize the importance of precise customization and planning to achieve optimal performance and compatibility with COST devices.

As part of our work, we prepared a clear and easy-to-follow guide for designing an acoustic signal tailored for localization purposes. However, further research and development are required to overcome the existing challenges and refine the system’s performance. Future work could focus on exploring advanced signal processing techniques, optimizing modulation schemes, and developing energy-efficient algorithms. In summary, our research contributes to the understanding and advancement of acoustic communication technologies for airborne channels. By addressing key technical aspects and conducting extensive measurements, we provide insights into future developments in this field. Our efforts aim to enhance communication capabilities in various applications, from consumer devices to industrial ones.

## Figures and Tables

**Figure 1 sensors-23-07852-f001:**
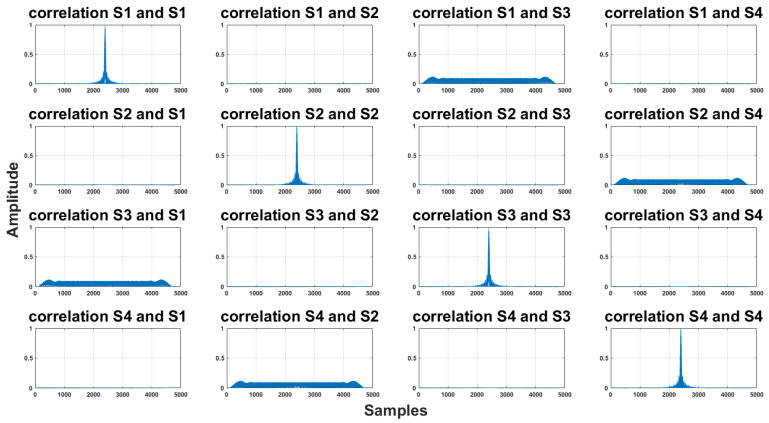
Correlation of the proposed signal at a sampling frequency of 48 kHz. In each row, there is a correlation of the signal with the other signals. The row number corresponds to the signal number, e.g., in the first row, it is the S1 signal, and for each column, the column number again represents the signal number, e.g., the second column represents S2. Auto-correlations for individual signals are located on the main diagonal.

**Figure 2 sensors-23-07852-f002:**
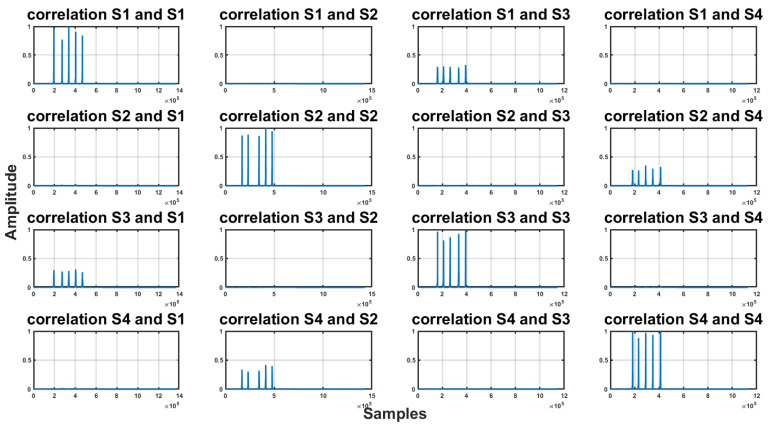
Correlation between signals on a recording of five repetitions of individual signals, with a sampling frequency of 48 kHz.

**Figure 3 sensors-23-07852-f003:**
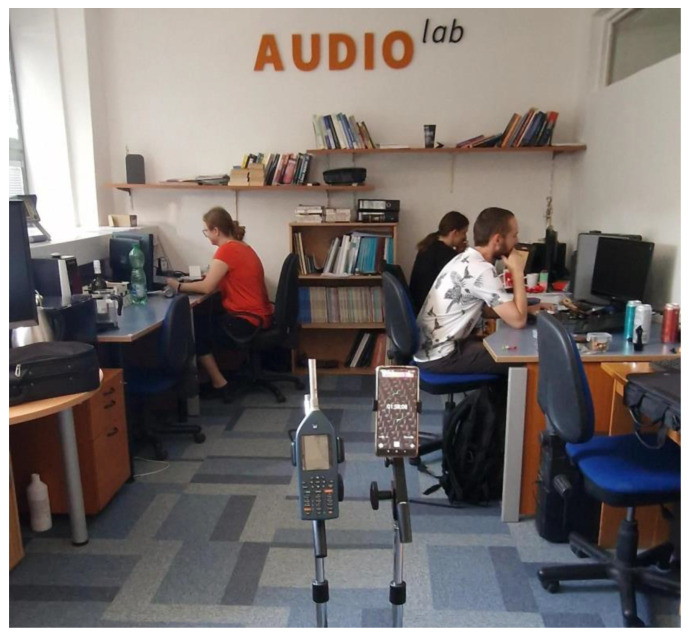
The measurement setup for background noise recording. Norsonic Nor140 on the left side and Xiaomi Redmi Note 10 Pro on the right side. Case office with carpet and people working on PCs.

**Figure 4 sensors-23-07852-f004:**
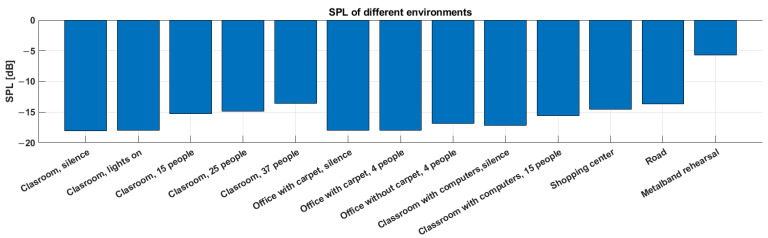
Background noise level reduced to the threshold of hearing for the selected frequency band 18–20 kHz under different conditions and rooms.

**Figure 5 sensors-23-07852-f005:**
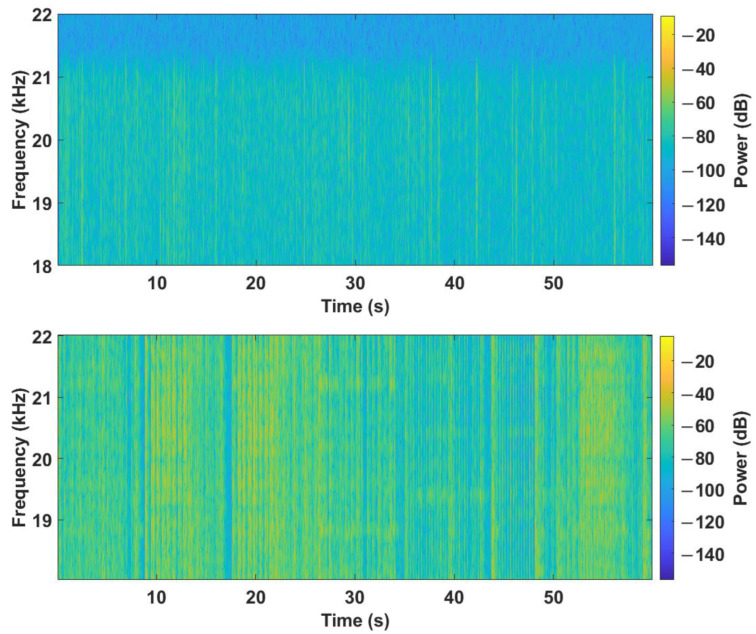
Sample spectrograms for the shopping center above and the metal band rehearsal below.

**Figure 6 sensors-23-07852-f006:**
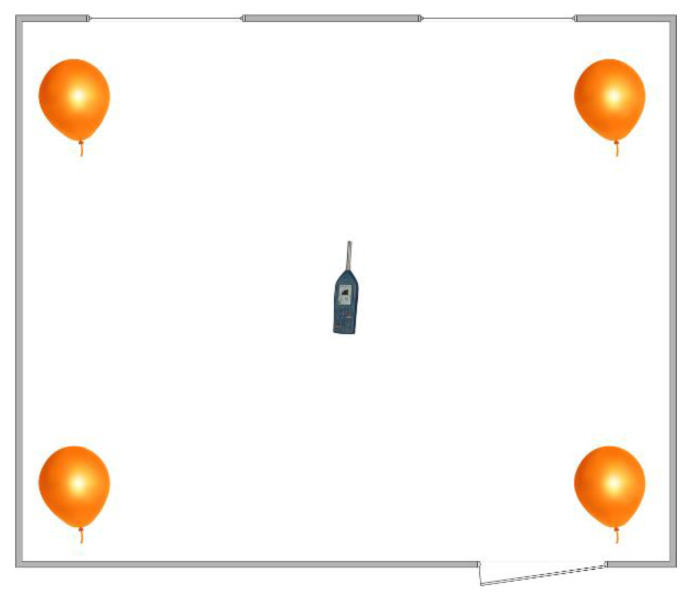
Design of the measurement system for the reverberation time.

**Figure 7 sensors-23-07852-f007:**
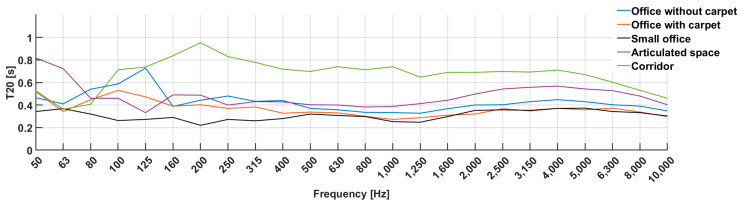
The T20 parameter depends on frequency and room.

**Figure 8 sensors-23-07852-f008:**
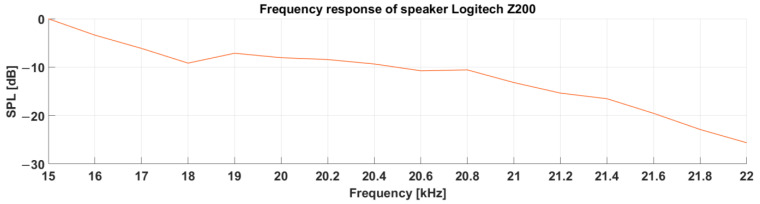
Frequency response of speaker Logitech Z200.

**Figure 9 sensors-23-07852-f009:**
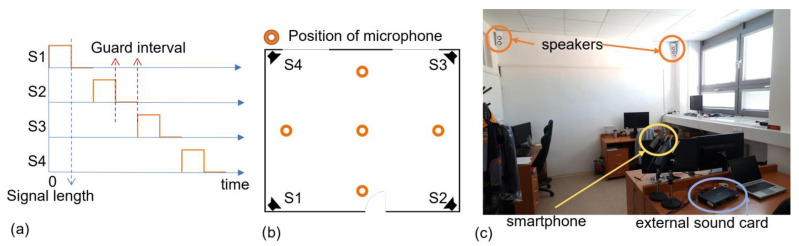
(**a**) Scheme of signal transmission; (**b**) location of the microphone during recording; (**c**) a photo of the measurement setup.

**Figure 10 sensors-23-07852-f010:**
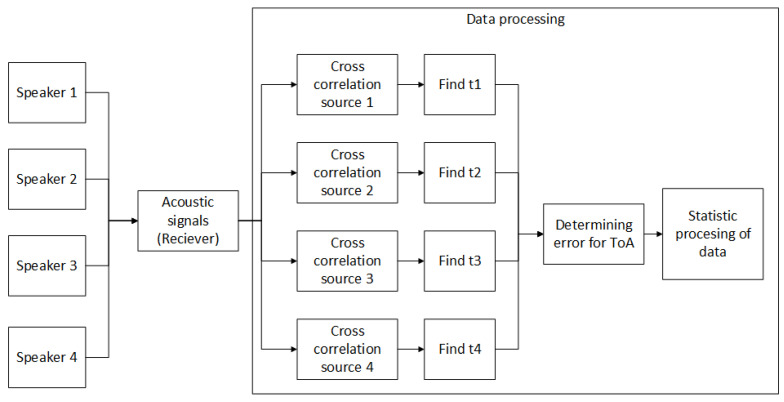
Process of individual measurement at each position. Statistic processing was performed for every room and every length of the guard interval.

**Figure 11 sensors-23-07852-f011:**
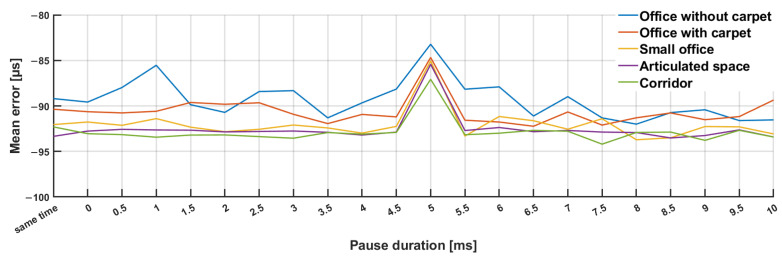
Mean ToA estimation error dependency on the size of the guard interval and the room type.

**Figure 12 sensors-23-07852-f012:**
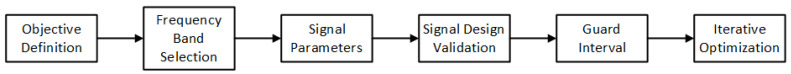
The steps for the design of an acoustic signal.

**Figure 13 sensors-23-07852-f013:**
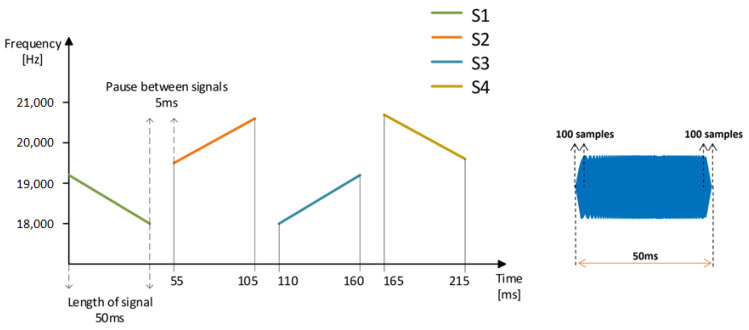
The final design of signals.

**Table 1 sensors-23-07852-t001:** An overview of the basic methods for determining the location of positioning systems.

Localization Methods		Measured Parameter
**Proximity method**	CMP (CoMmon proximity)	RSS
	CNP (CeNtroid proximity)	
	WEP (WEighted proximity)	
**Lateration**	Circular	Time
	Square	
	Hyperbolic	
**Angulation**		Angle
**Dead reckoning**		Several parameters are used, which are then used to calculate the position
**Fingerprinting**		RSS, background noise

**Table 2 sensors-23-07852-t002:** Description of designed individual signal from 4 sources.

Signal	Duration	Frequency [kHz]	Type
**S1**	50 ms	19.2–18	Down-chirp
**S2**	50 ms	19.4–20.6	Up-chirp
**S3**	50 ms	18–19.2	Up-chirp
**S4**	50 ms	20.6–19.4	Down-chirp

**Table 3 sensors-23-07852-t003:** Background noise level reduced to the threshold of hearing for the selected frequency band 18–20 kHz under different conditions and rooms.

Room	Number of People	Electronics Equipment	SPL [dB]
**Classroom**	0	off	−18.0
	0	lights	−18.0
	15	lights	−15.3
	25	lights	−14.9
	37	lights	−13.6
**Office with carpet**	0	off	−18.0
	4	4 PC, lights	−18.0
**Classroom with computers**	0	lights	−17.1
	15	server, 15 PC, lights	−15.6
**Office without carpet**	4	4 PC, lights	−16.8
**Shopping center**	>100	radio, lights	−14.6
**Road**	unspecified	no	−13.6
**Rehearsal metal band**	4	instruments, speakers, lights	−5.7

**Table 4 sensors-23-07852-t004:** The characteristics of selected rooms are described by their size and room equipment.

Room	Size [m]	Description
Office with carpet	9 × 4 × 2.8	Carpet, 4 PC, acoustic suspended ceilings, furniture
Office without carpet	6 × 3.6 × 2.8	Office, 4 PC, acoustic ceilings, tables, and chairs
Small office	4.08 × 2.80 × 2.8	0 PC, bookcase, cabinets, acoustic ceilings, 1 bare wall
Articulated space	6 × 6.5 × 2.8	Articulated space, cabinets, tables, acoustic ceilings, 2 PC
Corridor	2.09 × 10 × 2.6	Bare walls, acoustic ceilings, lights on

**Table 5 sensors-23-07852-t005:** Average T20 values from room impulse response for different frequencies.

Frequency	Locality				
	Office without Carpet	Office with Carpet	Small Office	Articulated Space	Corridor
**50 Hz**	0.465	0.52	0.3425	0.8175	0.5275
**63 Hz**	0.41	0.3425	0.3675	0.7225	0.3625
**80 Hz**	0.54	0.4475	0.32	0.46	0.4075
**100 Hz**	0.5875	0.53	0.2625	0.46	0.7125
**125 Hz**	0.7275	0.4725	0.2725	0.3325	0.7375
**160 Hz**	0.3875	0.39	0.29	0.49	0.8375
**200 Hz**	0.4425	0.4025	0.22	0.4875	0.9525
**250 Hz**	0.48	0.37	0.2725	0.4	0.83
**315 Hz**	0.4325	0.3825	0.26	0.43	0.7775
**400 Hz**	0.44	0.3275	0.28	0.4275	0.7175
**500 Hz**	0.37	0.3375	0.32	0.4025	0.6975
**630 Hz**	0.3575	0.33	0.3075	0.4	0.74
**800 Hz**	0.335	0.3	0.2975	0.3825	0.7125
**1 kHz**	0.3325	0.2725	0.2525	0.3875	0.74
**1.25 kHz**	0.3275	0.2875	0.2475	0.4125	0.6475
**1.6 kHz**	0.3675	0.31	0.2975	0.4425	0.69
**2 kHz**	0.4	0.32	0.3525	0.4975	0.69
**2.5 kHz**	0.4025	0.3675	0.3575	0.5425	0.6975
**3.15 kHz**	0.43	0.3475	0.3525	0.5575	0.6925
**4 kHz**	0.4475	0.37	0.37	0.5675	0.71
**5 kHz**	0.43	0.3575	0.3725	0.5425	0.67
**6.3 kHz**	0.4025	0.3725	0.3425	0.5275	0.6025
**8 kHz**	0.39	0.3375	0.3325	0.48	0.53
**10 kHz**	0.3475	0.2975	0.3025	0.4025	0.46
**A-netw**	0.3925	0.34	0.34	0.5	0.69
**Z-netw**	0.4125	0.35	0.35	0.4725	0.7075

**Table 6 sensors-23-07852-t006:** Equipment specifications during measurement.

Device		Specification:
**Reference device:**	Norsonic Nor140 with Nor1225 condenser microphone	Sensitivity: 50 mV/Pa Frequency response: 3.15 Hz to 22 kHz Measuring range: −10 to 137 dB
**Sound source:**	Logitech Z200	Frequency range: 80 Hz–20 kHz Power: 5 W RMS/10 W Control: Power/Volume, Tone Sound pressure level (SPL Max) > 88 dB
**External sound device (A/D and converter):**	Roland Rubix44	Sampling frequency:192 kHz
**Recording Software:**	For Android OS	WaveEditor
**Recording device:**	Xiaomi Redmi Note 10 Pro	Sampling frequency 48 kHz

**Table 7 sensors-23-07852-t007:** Comparison of the acoustic localization systems with a focus on signal design.

Work and Year	Frequency [kHz]	Duration of the Signal [ms]	Guard Interval [ms]	Signal Description
[47] 2023	18–30	50 pilot 150 whole	-	The pilot signal is an up-chirp of 18–30 kHz, a duration of 50 ms, followed by alternating up-chirps and down-chirps that signify 1 or 0. The overall signal duration is 150 ms, of which 100 ms is designated for the transmitter’s ID, which comprises 10 bits of information using CSS methods.
[48] 2022	20–23	45	-	Dual chirp with a duration of 45 ms, and the sampling frequency is set to 50.3 kHz. Custom-made hardware for transmitter and receiver.
[27] 2020	17–18 (preamble) 18–22 (message and ID)	40—preamble 30—message 30—ID	5	The acoustic signal is modulated by OCSS, using an overlap in the preamble with five tones that last 40 ms in the frequency band 17–18 kHz, with a difference of 200 Hz. Protection interval is 5 ms. The total length of the message is 110 ms.
[34] 2017	16—carrier frequency	63 bit	-	The carrier frequency is 16 kHz, the bandwidth is 8 kHz, it uses BPSK modulation, a 63-bit Kasami-coded signal, and the sampling frequency is 96 kHz. All transmitters were on one wall at different heights. They made measurements in one room with dimensions of 3 × 3 m. As receiver used iPad Air 2.
[49] 2021	15–21	50	100	Four chirp-signal patterns are defined as follows: 15–18, 18–15, 18–21, and 21–18 kHz. A total of 500 ms for the entire transmission of four signals together with a guard interval, i.e., 200 ms for signals and 300 for three guard intervals, each of which has a different duration.
[50] 2023	16–23	50	-	The proposed system develops a tightly coupled fusion platform of acoustic signal, BLE signal, and MEMS-IMU for the localization of commercial smartphones. A sampling frequency of 48 kHz, and custom-made emitters.
[51] 2022	12–22	20	-	Active acoustic sensing uses a 5–10 kHz signal with a duration of 3 ms. This signal uses a smartphone to measure the distance from the floor. Signals from emitters: 12–22 kHz and duration of 20 ms. Sampling frequency 48 kHz, custom-made emitters.
[52] 2022	15–22	50	200	S1 and S3 transmit an 18–15 kHz chirp signal at once, and S2 and S4 transmit the 19–22 kHz chirp signal at once after a 200 ms guard interval. It has a sampling frequency of 48 kHz.
Our design	18–20.6	50	5	A sampling frequency 48 kHz and four signals: 19.2–18, 19.4–20.6, 18–19.2, 20.6–19,2 kHz. A guard interval of 5 ms. Using COST devices.

## Data Availability

Not applicable.

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
