# Peer review of "Design of Acoustic Signal for Positioning of Smart Devices"

_sensors, 2023, doi:10.3390/s23187852_

Round 1

Reviewer 1 Report

In this work, the authors investigate several acoustic design parameters that play a role in the performance of acoustic positioning systems, such as the frequency range, signal duration and noise interference. More specifically, they show particular interest in the frequency range of 18 kHz to 22 kHz as existing hardware, such as smartphones, can be used. In general, three measurement campaigns were performed. (I) The authors measured the background noise levels in the frequency range 18 kHz to 22kHz for several rooms and conditions. (II) They measured the T20 reverberation time, again for several rooms and at different frequencies. (III) They investigated the pause time necessary between the transmitted signals by comparing the positioning accuracy in several rooms with different room characteristics.  

The introduction starts off with a lot of relevant background information, such as existing indoor positioning technologies, the fundamental groups of acoustic positioning systems based on frequency, etc. However, starting from line 87, comprehensive descriptions of existing acoustic positioning systems are provided. While it is certainly relevant to address their existence, it is not clear to me what the main take aways are for the reader. Moreover, it is also not clear whether other studies exist that already focus on this specific frequency range or maybe use smartphones for acoustic position estimations. 

The section 'Materials and methods' provides a clear overview of design parameters in acoustic positioning systems. However, ... 

1. More information should be provided for Equations (1) and (2) on page 5. More specifically, the transition from Equation (1) to (2) is not clear. Please also pay attention to the notation of certain parameters: R_gkgk(t) at line 215 and R_rgk(t) in Equation (2). 

2. The measurement description in section 2.6.1. is not clear. What exactly is measured by the equipment and what is obtained after post-processing in software. What does the measurement setup look like: can you provide some photos? What is the precision of the measurement setup for the reported parameters. For example, in the case of lines 335-336, a very low difference of 0.005 dB is reported. Can this difference be accounted for by the lights of is this the result of a poor measurement precision? What are the main take aways for the reader based on the reported background noise levels? What are the main conclusions for acoustic positioning systems? When will certain levels be problematic to such systems?

3. Why is the T20 parameter used in the measurements of paragraph 2.7.2, instead of the commonly used T60? Why are the specific boundaries of -5 dB and -25 dB used? Please explain the introduction of the special symbols in Equation (5)? Please explain why T60=T30=T20 at line 414. What main conclusions can be taken from Table 4 and Fig. 3 if the work targets positioning systems in the frequency range of 18 kHz to 22 kHz while the measurements only go up to 10 kHz? This is concern is also indicated by the authors at lines 450 and 451, however, why may one assume that the same trend will go on for higher frequencies?

4. The pause that is referred to in Section 2.7.3 is better known as the guard interval. Please use this term as it is widely known by the scientific community. Please explain how the guard band (or pause) is important to acoustic positioning systems. Have prior works already focused on this specific topic (if so, what are the main conclusions)?

5. The current structure of sections 2.7.3 and 2.8 makes it difficult to understand the measurement results presented in section 2.8.3. For example, the authors start off in section 2.7.3 with describing the measurement setup and the equipment specifications. However, it is not clear from the beginning which parameters will be used to assess a good guard band. It is only from line 522 at page 14 that the authors explain that the positioning accuracy will be used. Please explain exactly how to the position estimations are obtained from the measurements, how these are processed and how they ultimately lead to the measurement results presented in 2.8.3. A figure could help the reader to understand and evaluate this process. It is also not clear how the position estimations obtained at the different microphone locations (Fig. 4) are incorporated into the measurement results in 2.8.3. Furthermore, it is not clear what the contribution is of the measurements at the lower sample rate of 44.1 kHz in Fig. 7, especially because the results at 48 kHz in Fig. 8 show to be just fine.

6. lease explain the process of going from position estimations to the lognormal distribution in section 2.8.3, and thus ultimately to the measurement results in Fig. 9. The authors state that a guard band of 5 ms is optimal based on Fig. 9. However, how can this be concluded from the fact that µ only slightly increases towards the zero axis? Moreover, why does µ not further improve for larger guard bands? Is there an explanation why the guard band of 5ms is optimal compared to all other guard band durations?

Please improve the image resolutions of figures 3, 5, 6, 7, 8 and 9. Most of these figures will also be unreadable when printed out because the small font sizes. Please also provide more details in the captions of all figures. 

In some paragraphs there is a lot of repetition. Examples: lines 54 to 61 at page 2, lines 290 to 299 at page 7, lines 368 to 384 at page 9, lines 522 to 530, etc. 

In conclusion, the paper does not present new insights that further progress the state-of-the-art of acoustic positioning systems. While the study addresses relevant performance parameters of such systems, no new and irrefutable design guidelines are concluded from the measurement campaigns. Furthermore, more relevant work on acoustic positioning using smartphones should be included in the literature study. 

Please pay attention to missing 'articles' (e.g., a, an, the) throughout the paper text. 

Author Response

Reviewer#1, Concern # 1: The introduction starts off with a lot of relevant background information, such as existing indoor positioning technologies, the fundamental groups of acoustic positioning systems based on frequency, etc. However, starting from line 87, a comprehensive description of existing acoustic positioning systems are provided. While it is certainly relevant to address their existence, it is not clear to me what the main take aways are for the reader. Moreover, it is also not clear whether other studies exist that already focus on this specific frequency range or maybe use smartphones for acoustic position estimations.

Author response:  Thank you for the comment.

Author action: The manuscript was updated by the part from line 87 was reduced, and a table for comparison with relevant existing systems was added.  

Reviewer#1, Concern # 2: The section 'Materials and methods' provides a clear overview of design parameters in acoustic positioning systems. However, ...

  1. More information should be provided for Equations (1) and (2) on page 5. More specifically, the transition from Equation (1) to (2) is not clear. Please also pay attention to the notation of certain parameters: R_gkgk(t) at line 215 and R_rgk(t) in Equation (2).

Author response:  Thank you for the comment. Equation (1) describes the received signal in MS, which is changed from the transmitted signal by passing through the communication channel. Equation (2) describes the output of the receiver, which is formed by correlating r(t) with all signal codes.

Author action: We updated the manuscript and included the explanation of these equations.

  1. The measurement description in section 2.6.1. is not clear. What exactly is measured by the equipment and what is obtained after post-processing in software. What does the measurement setup look like: can you provide some photos? What is the precision of the measurement setup for the reported parameters. For example, in the case of lines 335-336, a very low difference of 0.005 dB is reported. Can this difference be accounted for by the lights of is this the result of a poor measurement precision? What are the main take aways for the reader based on the reported background noise levels? What are the main conclusions for acoustic positioning systems? When will certain levels be problematic to such systems?

Author response:  Thank you for the comment. The section 2.6.1.  describes the measurement of background noise in the selected frequency band of 18-20 kHz, as we want to use our acoustic signal for localization in this band. After postprocessing in software, the SPL [dB] value of background noise is obtained. Section 2.6.1 has been revised for enhanced clarity. Please find the updated description below. Additionally, a photo of the measurement setup is included for better understanding.

The precision of the measurement base on the specification of the Norsonic Nor140 is gain accuracy at 1 kHz: ±0.2 dB, frequency response re. 1 kHz: ±0.5 dB for 20 Hz < f< 20 kHz.  Norsonic Nor140 is a class 1 analyser, class 1 means that it is professional equipment for acoustic analysing. So, as you rightly pointed out, such a small value corresponds to measurement inaccuracy. The corresponding part in the article was edited along with the addition of information about the accuracy of the device.

The main take aways from this measurement is, that SPL in this frequency range is low, so interference with the acoustic signal used for the estimation of the position is low, based on the SPL of the acoustic signal setup. The problem comes with small SNR between background noise and acoustic signal for localization.

Author action:  Whole sections 2.6. was updated according to the comments from reviewers.

  1. Why is the T20 parameter used in the measurement of paragraph 2.7.2, instead of the commonly used T60? Why are the specific boundaries of -5 dB and -25 dB used? Please explain the introduction of the special symbols in Equation (5)? Please explain why T60=T30=T20 at line 414. What main conclusions can be taken from Table 4 and Fig. 3 if the work targets positioning systems in the frequency range of 18 kHz to 22 kHz while the measurements only go up to 10 kHz? This is concern is also indicated by the authors at lines 450 and 451, however, why may one assume that the same trend will go on for higher frequencies?

Author response:  Thank you for the comment. Most relevant for measuring reverberation time is the initial 20-30dB of the 60 dB sound decay. In terms of sound degradation, this is the part that the human ear can most easily perceive. In most contexts, a decline of 60dB is difficult to accomplish. T20 or T30 can be used and extrapolated to determine the reverberation time as described in [1] and [2]. The Nor140 calculate the T20 value which is normalized to the required 60 dB decay time.

[1] F. A. Everest and K. C. Pohlmann, Master Handbook of Acoustics. New York: The McGraw-Hill Companies, Inc., cop. 2015.

[2] ‘ISO 3382-1:2009(en), Acoustics — Measurement of room acoustic parameters — Part 1: Performance spaces’. https://www.iso.org/obp/ui/#iso:std:iso:3382:-1:ed-1:v1:en (accessed May 15, 2023).

For the T20 parameter, specific boundaries of -5 dB and -25 dB are given by [2]. The equation (5) symbols mean the time at which the decay curve first reaches 5 dB and 25 dB below the initial level. In simpler terms, ξ represents the SPL value (-5 dB or -25 dB).

T60=T30=T20 applies to linear systems, it is not necessary for an explanation of our measurement and this line was removed from the paper.

The measurements of T20 were only up to a value of 10 kHz, as our Norsonic Nor140 equipment is limited to this value, and various software for determining these parameters are normally limited to 8 kHz or up to 10 kHz. These limits are because higher frequencies are uninteresting from an acoustic point of view (music, speech). The assumption for the decreasing nature of the curve is based on the properties of the acoustic signal i.e., attenuation by propagation, higher frequencies are attenuated faster. Sound attenuation coefficient ISO 9613-1 and ISO 9613-2 (also up to 10kHz).  For value 18 kHz we obtain A=0.4992 [dB/m], for 20 kHz it is A=0.5783 [dB/m].

Author action: We updated the manuscript by adding an explanation for T20 and equation (5).

  1. The pause that is referred to in Section 2.7.3 is better known as the guard interval. Please use this term as it is widely known by the scientific community. Please explain how the guard band (or pause) is important to acoustic positioning systems. Have prior works already focused on this specific topic (if so, what are the main conclusions)?

Author response:  Thank you for the comment. The guard interval is required to compensate ISI, which affects the correlation maxima and thus introducing errors in the ToA estimates. According to our literature review, other teams didn’t focus on this specific topic for indoor positioning using acoustic signal. To investigate the impact of various room characteristics on the guard interval between signals in acoustic positioning systems, we designed and implemented a measurement setup. The primary goal of this experiment was to understand how room properties influence the necessary guard interval between successive signals to achieve accurate positioning. The guard interval is essential in compensating for ISI, which affects the correlation maxima, making it challenging to precisely determine the ToA of the transmitted signal and introduce errors. Measurements were performed in different rooms with diverse acoustic properties, representing a range of sizes, shapes, and materials, as described in section 2.7.2. The emitters emitted signals at specific intervals, and the receivers recorded it. During the measurements, the guard interval between signals was systematically varied, starting from a minimal value and gradually increasing it. By analyzing the received signals, we assessed how room parameters, such as size, shape, surface materials, and reverberation time, affect ToA based on the size of the guard interval between signals. These factors can influence signal propagation, reflections, and reverberation within the room.

Author action: We updated the manuscript by adding an explanation for measurement different guard intervals.

  1. The current structure of sections 2.7.3 and 2.8 makes it difficult to understand the measurement results presented in section 2.8.3. For example, the authors start off in section 2.7.3 with describing the measurement setup and the equipment specifications. However, it is not clear from the beginning which parameters will be used to assess a good guard band. It is only from line 522 at page 14 that the authors explain that the positioning accuracy will be used. Please explain exactly how to the position estimations are obtained from the measurements, how these are processed and how they ultimately lead to the measurement results presented in 2.8.3. A figure could help the reader to understand and evaluate this process. It is also not clear how the position estimations obtained at the different microphone locations (Fig. 4) are incorporated into the measurement results in 2.8.3. Furthermore, it is not clear what the contribution is of the measurements at the lower sample rate of 44.1 kHz in Fig. 7, especially because the results at 48 kHz in Fig. 8 show to be just fine.

Author response:  Thank you for the comment. We rearranged this part to make it clear and easy to follow.

Initially, a sampling frequency of 44.1 kHz was selected for recording the signal, but during the real measurement, we found that using this sampling frequency, individual acoustic signal sources cannot be distinguished, so this frequency was subsequently increased to 48 kHz. This is also shown in fig. 6-8. We rearranged the part 2.7. and 2.8.

Author action: We updated the manuscript with rearranged the part 2.7. and 2.8.

  1. Please explain the process of going from position estimations to the lognormal distribution in section 2.8.3, and thus ultimately to the measurement results in Fig. 9. The authors state that a guard band of 5 ms is optimal based on Fig. 9. However, how can this be concluded from the fact that μ only slightly increases towards the zero axis? Moreover, why does μ not further improve for larger guard bands? Is there an explanation why the guard band of 5ms is optimal compared to all other guard band durations?

Author response: Thank you for the comment. In section 2.8.3 impact of the guard interval on accuracy od ToA measurements was evaluated. The ToA values are data that represent input of the localization system. The results of the measurements were analyzed and based on this analysis the distribution of ToA estimation error had lognormal distribution.  

Based on the correlation, ToA was obtained from the recordings, and errors in ToA determination were subsequently calculated. Statistical processing of these error values was performed in each room and for each guard interval length using the MATLAB environment and statistical toolbox. The goal was to analyse the error distribution and determine its parameters. By examining the data, we aimed to identify the guard interval length that minimizes the error in ToA estimation, providing valuable insights into optimizing the acoustic positioning system for accurate and reliable performance. For more clarity see Fig. 4.

Fig. 4. Process of individual measurement at each position. Statistic processing were done for every room and every length of guard interval.

The increase of μ towards the zero axis indicates that a duration of 5 ms results in the smallest error when determining Time of Arrival (ToA) with the chosen signal design.

Author action: We updated the manuscript by adding an explanation for data processing.

Reviewer#1, Concern # 3: Please improve the image resolutions of figures 3, 5, 6, 7, 8 and 9. Most of these figures will also be unreadable when printed out because the small font sizes. Please also provide more details in the captions of all figures. In some paragraphs there is a lot of repetition. Examples: lines 54 to 61 at page 2, lines 290 to 299 at page 7, lines 368 to 384 at page 9, lines 522 to 530, etc.

Author response:  Thank you for the comment. Repetition was reduced and the quality of figures was improved.

Author action: All figures' font size was enlarged and details in the caption were supplemented. Repetition was reduced.

Reviewer#1, Concern # 4: In conclusion, the paper does not present new insights that further progress the state-of-the-art of acoustic positioning systems. While the study addresses relevant performance parameters of such systems, no new and irrefutable design guidelines are concluded from the measurement campaigns. Furthermore, more relevant work on acoustic positioning using smartphones should be included in the literature study.

Author response:  Thank you for the comment.

Author action: We have updated the paper according to the valuable comments provided by the reviewer. We believe the paper is now much clearer and more valuable.

More detailed response can be found in the attached pdf file.

Reviewer 2 Report

The paper needs to include (crucial) technical specifications and noise characterization in the specified frequency range, limiting its potential contribution to innovative localization systems. The study needs to provide a robust analysis of signal design and performance, as it needs experimental results and comparisons with existing methods. Merely mentioning previous research without providing specific references or supporting evidence undermines the credibility of the findings.

Moreover, the authors' justification for signal length and pause duration must adequately explain the reasoning behind their choices. The lack of detailed methodology and measurement procedures raises doubts about the reliability and reproducibility of the results.

The frequency band selection appears arbitrary and thus needs a comprehensive analysis. The absence of clear criteria for band selection and insufficient details about the measurement process undermine the validity of the conclusions drawn. Suggest adding and discussing references in related domains: 10.1016/j.measurement.2017.11.046; 10.1080/10447318.2023.2209836

Without concrete experimental evidence and thorough analysis, the proposed acoustic communication system remains unsubstantiated and unconvincing. Revisions, including rigorous analysis and clear methodological explanations, are necessary to make this paper publishable in this journal.

na

Author Response

Reviewer#2, Concern # 1: The paper needs to include (crucial) technical specifications and noise characterization in the specified frequency range, limiting its potential contribution to innovative localization systems. The study needs to provide a robust analysis of signal design and performance, as it needs experimental results and comparisons with existing methods. Merely mentioning previous research without providing specific references or supporting evidence undermines the credibility of the findings.

Author response:  Thank you for the comment. The crucial technical specification was included in the paper. The characteristics of noise in the given frequency range does depend on the characteristics of the environment, however, according to the measurements results provided in the section 2.6.1. the noise levels are extremely low. Moreover, spectrogram of the noise characteristics was included in the updated paper.

Author action: We have updated the paper according to the valuable comments provided by the reviewer. We believe the paper is now much clearer and more valuable.

Reviewer#2, Concern # 2: Moreover, the authors' justification for signal length and pause duration must adequately explain the reasoning behind their choices. The lack of detailed methodology and measurement procedures raises doubts about the reliability and reproducibility of the results.

Author response:  Thank you for the comment. The frequency range of 18 - 20.6 kHz was selected for two main reasons. First, it falls above the 18 kHz threshold [1], [2], making the acoustic signal inaudible for most of the hu-mans. Second, it aligns with the frequency characteristics of the microphones, as explained in paragraph 2.1. and detailed in our publication [3]. This frequency range pro-vides us with 2.6 kHz of bandwidth, we need to create four distinct signals, each designed to be easily distinguishable through correlation. With these considerations, we ensure efficient signal separation and accurate identification of the acoustic signal sources. We set the signal length to 50 ms based on the results of the authors in [4], who found that a signal with a length of 40 ms or more has an 80% higher rate efficiency (number of successful experiments/total number of experiments) and a 0.05 m lower average error of distance estimates. We were limited by the bandwidth, which is 18-20.6 kHz, and the need to create four different signals. Furthermore, the signal frequency is constrained by the array spacing, which must fulfil the equation length of signal < speed of sound/2frequency band, which means that our length of signal has to bee shorter than 0.066 s.

The measurement description was enhanced by including additional explanatory figures for better clarity and understanding.

[1] K. Ashihara, ‘Threshold of hearing for pure tones between 16 and 30 kHz’, The Journal of the Acoustical Society of America, vol. 120, no. 5, pp. 3245–3245, Nov. 2006, doi: 10.1121/1.4788280.

[2] P. G. Stelmachowicz, K. A. Beauchaine, A. Kalberer, and W. Jesteadt, ‘Normative thresholds in the 8‐ to 20‐kHz range as a function of age’, The Journal of the Acoustical Society of America, vol. 86, no. 4, pp. 1384–1391, Oct. 1989, doi: 10.1121/1.398698.

[3] V. Hromadová, P. Kasák, R. Jarina, and P. Brída, ‘Frequency Response of Smartphones at the Upper Limit of the Audible Range’, in 2022 ELEKTRO (ELEKTRO), May 2022, pp. 1–5. doi: 10.1109/ELEKTRO53996.2022.9803475.

[4] G. Li, L. Zhang, F. Lin, M. Chen, and Z. Wang, ‘SAILoc: A novel acoustic single array system for indoor localization’, in 2017 9th International Conference on Wireless Communications and Signal Processing (WCSP), Oct. 2017, pp. 1–6. doi: 10.1109/WCSP.2017.8171099.

Author action: We have updated the paper supplemented with explanation and reasons behind our choices.

Reviewer#2, Concern # 3: The frequency band selection appears arbitrary and thus needs a comprehensive analysis. The absence of clear criteria for band selection and insufficient details about the measurement process undermine the validity of the conclusions drawn. Suggest adding and discussing references in related domains: 10.1016/j.measurement.2017.11.046; 10.1080/10447318.2023.2209836

Author response:  Thank you for the comment. We hope that the previous answer clarified our decisions for the choice of frequency band. We have added the suggested references to the article.

Author action: We updated the manuscript by adding an explanation and suggested references to the article.

Reviewer#2, Concern # 4: Without concrete experimental evidence and thorough analysis, the proposed acoustic communication system remains unsubstantiated and unconvincing. Revisions, including rigorous analysis and clear methodological explanations, are necessary to make this paper publishable in this journal.

Author response:  Thank you for the comment.

Author action: We have updated the paper according to the valuable comments provided by the reviewer. We believe the paper is now much clearer and more valuable.

Reviewer 3 Report

sensors-2520710

Title: Design of Acoustic Signal for Positioning of Smart Devices

The manuscript is difficult to follow. Some points need to be known.

-Introduction is looking too big. Please try to reduce it.

-What Table 3 shows?

-It will be good to include the real picture of the experimental setup showing positioning emitters and receivers.

-Please briefly discuss the real-time application of Acoustic Signals for the Positioning of Smart Devices.

-Have authors tested the signal with different temperatures and humidity?

-Figures 6 to 8 are not clear. Please modify.

- The abstract and conclusion should be appropriately written, emphasizing the novelty of the work.

-It is better to list a comparison table to compare results with previous work.

-Please add some latest references.

Minor editing of English language required.

Author Response

Reviewer#3, Concern # 1: Introduction is looking too big. Please try to reduce it.

Author response:  Thank you for the suggestion.

Author action: We reduced the introduction.

Reviewer#3, Concern # 2: What Table 3 shows?

Author response:  Table 3 provides a description of the rooms chosen for measuring the room's reverberation time. Every room is described by its size and room equipment. We want to establish their acoustical properties and compare them, and then utilize the same rooms to analyze the impact of the guard interval (or pause) between signals on the ToA estimation error.

Author action: We added the description of the table, we also described it more in the text.

Reviewer#3, Concern # 3: It will be good to include the real picture of the experimental setup showing positioning emitters and receivers.

Author response:  Thank you for the suggestion. The real photo from the measurement was added to paper.

Author action: We included a real picture from the experiment setup for a receiver and emitters.

Reviewer#3, Concern # 4: Please briefly discuss the real-time application of Acoustic Signals for the Positioning of Smart Devices.

Author response:  The real-time application of acoustic signals for the positioning of smart devices holds significant potential in various fields, offering a range of practical and innovative applications. Acoustic positioning utilizes sound signals for determining the location of smart devices in indoor or outdoor environments, making it a valuable tool for location-based services and context-aware applications. Here are some key aspects of its real-time application:

  • Indoor Localization: Acoustic signals can be used for precise indoor localization of smart devices in environments where GPS signals may be unreliable or unavailable. This enables seamless navigation and tracking within buildings, such as shopping malls, airports, hospitals, and museums.
  • Asset Tracking: In industrial settings, acoustic positioning can be employed to track assets, equipment, or inventory. This real-time tracking helps optimize logistics, prevent theft, and improve operational efficiency.
  • Context-Aware Services: Acoustic signals can serve as contextual cues for smart devices, providing relevant information based on their location. For instance, museums can offer audio guides triggered by the user's proximity to exhibits.
  • Emergency Response: In emergencies, acoustic positioning can be utilized for tracking people's locations in real-time, aiding rescue efforts and ensuring safety.
  • Smart Home Automation: By integrating acoustic positioning into smart homes, devices can adjust settings based on the occupants' locations, optimizing energy consumption, and enhancing user experience.
  • Augmented Reality (AR): AR applications can leverage acoustic positioning to overlay virtual elements onto the real world accurately, enriching user experiences in gaming, education, or entertainment.
  • Proximity-Based Interaction: Acoustic signals enable proximity-based interactions between smart devices, allowing seamless data transfer, sharing, and communication when devices are in close vicinity.
  • Location-Based Marketing: Businesses can employ acoustic positioning for targeted location-based marketing, delivering personalized offers and promotions to customers based on their proximity to specific areas.

Overall, the real-time application of acoustic signals for smart device positioning opens new opportunities for enhancing user experiences, optimizing processes, and enabling a wide range of innovative services across various domains. As technology advances, acoustic positioning will likely continue to evolve and find even more diverse applications in the future.

Reviewer#3, Concern # 5: Have authors tested the signal with different temperatures and humidity?

Author response:  We didn’t test the signal with different temperatures and humidity. These parameters affect the propagation speed of the acoustic signal. The dependence of the propagation speed of the acoustic signal in the interior on the change in temperature and humidity can be described as follows. A change in temperature of 1°C means a change in velocity of 0.606 ms-1 for dry air, as can be seen from the following equation (1). Where c represents the propagation speed of the acoustic signal, T represents the thermodynamic temperature [K], κ is Poisson's constant, R is the universal gas constant, and M is the molecular weight of the environment. This relation can be used at a frequency greater than 1 Hz (events in the gas are adiabatic after the passage of the acoustic signal). The effect of humidity on the rate is based on the average molecular weight, which decreases, while Poisson's constant also changes [1].

c=√(κRT/M)

(1)

In the indoor environment, values of temperature and humidity do not fluctuate much over time. Moreover, we would like to note that 5°C change in temperature (which is rather dramatic in the indoor environment) would represent a 1% change in speed, which would have relatively small effect on resulting accuracy of the ToA measurements and localisation system.  

[1] A. Nowoświat, ‘Impact of Temperature and Relative Humidity on Reverberation Time in a Reverberation Room’, Buildings, vol. 12, no. 8, Art. no. 8, Aug. 2022, doi: 10.3390/buildings12081282.

Reviewer#3, Concern # 6: Figures 6 to 8 are not clear. Please modify.

Author response:  Thank you for the comment.

Author action: All figures' font size was enlarged and caption was supplemented for more clarity.

Reviewer#3, Concern # 7: The abstract and conclusion should be appropriately written, emphasizing the novelty of the work.

Author response:  Thank you for the suggestion. Based on the comment the abstract and conclusion were rewritten.

Author action: We updated the manuscript with a rewritten abstract and conclusion.

Reviewer#3, Concern # 8: It is better to list a comparison table to compare results with previous work.

Author response:  Thank you for the comment.

Author action: We added a table for comparison with previous work.

Reviewer#3, Concern # 9: Please add some latest references.

Author response:  Thank you for the comment. We added some latest references.

Author action: We added the latest references into the manuscript.

Round 2

Reviewer 1 Report

- Please re-read the article thoroughly for spelling or other visual errors. For example: 'squere' in Table 1, 'where good results were' at line 188, the equation at line 254, sudden spacing between lines 550 and 552.

- While the measurement setup and strategy in 2.7.4 is explained in more detail (compared to revision no. 1) it is still not clear why a guard band of 5 ms is most optimal, as is presented in Figure 11. This comment was not addressed by the authors in revision 1. How can this be concluded from the measurements based on the fact that µ only slightly increases towards the zero axis? Why does µ not improve for larger guard bands, or at least be as good as in the case of 5 ms, since they affect each other less and less? Can it be expected that the same applies for other measurements setups: if it is not universal, can it be a design guideline for other systems? Moreover, the difference in mean error between the peak at 5ms and other pause durations is only in the order of 1µs: at the speed of sound (343 m/s) this result in an error smaller than 1mm, which is extremely low for acoustic ranging/positioning systems in indoor environments? Please re-evaluate the results of the measurement campaign.

-

Author Response

Reviewer#1, Concern # 1: Please re-read the article thoroughly for spelling or other visual errors. For example: 'squere' in Table 1, 'where good results were' at line 188, the equation at line 254, sudden spacing between lines 550 and 552.

Author response: Thank you for the comment. Errors were fixed.

Author action: The manuscript was revised to correct typos and enhance the English.

Reviewer#1, Concern # 2: While the measurement setup and strategy in 2.7.4 is explained in more detail (compared to revision no. 1) it is still not clear why a guard band of 5 ms is most optimal, as is presented in Figure 11. This comment was not addressed by the authors in revision 1. How can this be concluded from the measurements based on the fact that μ only slightly increases towards the zero axis? Why does μ not improve for larger guard bands, or at least be as good as in the case of 5 ms, since they affect each other less and less? Can it be expected that the same applies for other measurements setups: if it is not universal, can it be a design guideline for other systems? Moreover, the difference in mean error between the peak at 5ms and other pause durations is only in the order of 1μs: at the speed of sound (343 m/s) this result in an error smaller than 1mm, which is extremely low for acoustic ranging/positioning systems in indoor environments? Please re-evaluate the results of the measurement campaign.

Author response: Thank you for the comment. We have checked the data and re-evaluated the results of the measurements. Indeed we have found there was a mistake in the plotting of the data, where time values were incorrectly converted to us. We corrected the script and provided updated figure in the updated manuscript.

Measurements and evaluation of different signal designs and hardware configurations can help gain a comprehensive understanding of the optimal guard interval and signal parameters. However, such a comprehensive measurement campaign is beyond the scope of this manuscript.

We added a new Section 2.8 for explaining the steps for the design of an acoustic signal for localization purposes in general.

Author action: We updated the manuscript and included the explanation:

Measurements show that the value of μ decreases slightly towards the zero axis for a guard interval equal to 5 ms. This indicates that a guard interval of 5 ms represents the smallest error in ToA estimation for the selected signal design. We expected the course of the curve to show an improving trend with an increasing protection interval. The data shows that the room does not have a great influence on the accuracy of Time of Arrival (ToA) estimates. Errors in ToA of around 10 microseconds, with the speed of sound (c) being 343 m/s, result in position estimation errors on the order of centimetres.

The optimal guard interval length seems to depend on the shaping of the audio signal as well as the used frequency band and sampling rate. The outcome is affected by the aliasing and artefacts resulting from the sampling process.

Therefore, the results of this specific measurement may not be universal and may depend on specific signal design and used hardware. It is necessary to consider these factors and perform thorough measurements for any other specific signal designs.

2.8. Guidelines for Designing Acoustic Signals for Localization Purposes

The steps for the design of an acoustic signal for localization purposes, in general, can be seen in Fig. 12.

Fig. 12. The steps for the design of an acoustic signal

Individual design steps:

  1. Objective Definition: Clearly define the objective of the acoustic localization system. Identify the specific localization requirements, such as accuracy, and range. Consider environmental conditions such as room dimensions, materials, and ambient noise. Considers the limitations and compatibility with the hardware and software capabilities of the target devices, such as smartphones.
  2. Frequency Band Selection: Choose an appropriate frequency band for the acoustic signal design. Consider the trade-offs between higher frequencies (better resolution but shorter range) and lower frequencies (lower resolution but longer range). Consider target device limitations.
  3. Signal Parameters: Decide on the type of acoustic signal to be used, such as continuous wave, chirp signals, or coded signals (e.g., pseudo-noise codes). Determine the essential signal parameters, including signal duration, modulation scheme (e.g., BPSK, FSK), and coding method (e.g., Kasami codes). Choose a suitable sampling frequency for signal processing. Higher sampling rates provide better resolution but may require more processing power.
  4. Guard Interval: Calculate and set an appropriate guard interval between consecutive signals to avoid interference and ensure accurate signal detection. This interval should consider the signal's duration, room acoustics, and potential signal reflections.
  5. Signal Design Validation: Use simulation or analytical tools to validate the design and ensure it meets the desired localization requirements. Test the signal in a controlled environment to verify its performance. Conduct real-world experiments in different environments to evaluate the acoustic signal's effectiveness and accuracy for localization.
  6. Iterative Optimization: Continuously refine and optimize the signal design based on experimental results and feedback from real-world tests.

In summary, designing an effective acoustic signal for localization requires a systematic approach involving clear objective definition, careful frequency band selection, determination of signal parameters, appropriate guard interval calculation, thorough signal design validation through simulations and experiments, and continuous iterative optimization. By following these guidelines, developers can create acoustic signals that meet specific localization requirements and are compatible with the hardware and software capabilities of target devices.

Reviewer 3 Report

sensors-2520710

Design of Acoustic Signal for Positioning of Smart Devices

Thank you for allowing me to revise resubmitted manuscript titled " Design of Acoustic Signal for Positioning of Smart Devices" I believe the submitted manuscript and presented work is suitable for publishing in Sensors.

Minor editing of English language required

Author Response

Reviewer#3, Concern # 1: Thank you for allowing me to revise resubmitted manuscript titled " Design of Acoustic Signal for Positioning of SmartDevices" I believe the submitted manuscript and presented work is suitable for publishing in Sensors. Minor editing of English language required.

Author response: Thank you for the comment.

Author action: The manuscript was revised to correct typos and enhance the English.
